# Ground-Dwelling Arthropod Community Responses to Recent and Repeated Wildfires in Conifer Forests of Northern New Mexico, USA

**Scott Ferrenberg** [1],\*, **Philipp Wickey** [2] and **Jonathan D. Coop** [2]

1   Department of Biology, New Mexico State University, Las Cruces, NM 88003, USA
2   School of Environment and Sustainability, Western Colorado University, Gunnison, CO 81231, USA
\*   Correspondence: ferrenbe@nmsu.edu

**Abstract:** The increasing frequency and severity of wildfires in semi-arid conifer forests as a result of global change pressures has raised concern over potential impacts on biodiversity. Ground-dwelling arthropod communities represent a substantial portion of diversity in conifer forests, and could be particularly impacted by wildfires. In addition to direct mortality, wildfires can affect ground-dwelling arthropods by altering understory characteristics and associated deterministic community assembly processes (e.g., environmental sorting). Alternatively, disturbances have been reported to increase the importance of stochastic community assembly processes (e.g., probabilistic dispersal and colonization rates). Utilizing pitfall traps to capture ground-dwelling arthropods within forest stands that were burned by one or two wildfires since 1996 in the Jemez Mountains of northern New Mexico, United States (USA), we examined the potential influences of deterministic versus stochastic processes on the assembly of these diverse understory communities. Based on family-level and genera-level arthropod identifications, we found that the multivariate community structures differed among the four fire groups surveyed, and were significantly influenced by the quantities of duff, litter, and coarse woody debris, in addition to tree basal area and graminoid cover. Taxon diversity was positively related to duff quantities, while taxon turnover was positively linked to exposed-rock cover and the number of logs on the ground. Despite the significant effects of these understory properties on the arthropod community structure, a combination of null modeling and metacommunity analysis revealed that both deterministic and stochastic processes shape the ground-dwelling arthropod communities in this system. However, the relative influence of these processes as a function of time since the wildfires or the number of recent wildfires was not generalizable across the fire groups. Given that different assembly processes shaped arthropod communities among locations that had experienced similar disturbances over time, increased efforts to understand the processes governing arthropod community assembly following disturbance is required in this wildfire-prone landscape.

**Keywords:** arthropods; bandelier national monument; cerro grande fire; ground-dwelling community; duff; jemez mountains; las conchas fire; litter; reburn; understory vegetation

## 1. Introduction

Fire is an important ecological process globally [1]. Human activities have greatly altered fire regimes via suppression [2], ignitions [3], and changes in the quantity and arrangement of fuels [4]. Additionally, warmer and drier conditions associated with climate change have increased fire activity in many regions [5–8]. Wildfires resulting from these pressures can strongly affect biodiversity [9,10]. Thus, as wildfires become more frequent and severe as a consequence of global change [1], understanding their impact on the biotic communities of forest ecosystems remains an important focus.

Ground-dwelling arthropod communities represent a substantial proportion of biodiversity in forest ecosystems [11]. These communities have been reported to be sensitive to alterations in vegetation and litter cover from various forest disturbances, ranging from severe wildfires to relatively minor manipulations of coarse woody debris [12–21]. Thus, fires may affect ground-dwelling arthropods through direct mortality and/or via impacts on soil and litter/duff layers, understory and overstory vegetation composition, the quantity and arrangement of woody debris, and changes in microclimate [12,22–25]. While numerous studies have found that fires decrease arthropod density, e.g., [12,24], responses at a community level can seem idiosyncratic, and some taxa appear to be resistant to fire disturbance, while others are highly sensitive [26–30]. Given the apparent context-dependence of responses to fire disturbances across arthropod taxa and ecosystems, a focus on the inferring processes governing arthropod community assembly could improve predictions of the disturbance impacts on these diverse communities.

Understanding the processes governing the assembly of biotic communities is a longstanding goal in ecology. A great deal of work on this topic has focused on the role of deterministic processes such as environmental filtering or niche-sorting and antagonistic and synergistic species interactions (e.g., [31,32]). At the same time, stochastic processes linked to the probability of dispersal, colonization, and local extinction can also be key drivers of community structure [33–37]. More recently, simultaneous influences of deterministic and stochastic processes in the assembly of biotic communities, including numerous arthropod assemblages, have been reported (e.g., [13,38–43]).

Numerous factors are hypothesized to determine the relative influence of deterministic and stochastic assembly processes in biotic communities—e.g., ecosystem productivity, regional biodiversity and dispersal rates, habitat connectivity, species' interactions, and disturbances [40,43,44]. Of these factors, disturbances have been reported to increase the relative influence of stochastic processes in the short-term, with an increase in deterministic processes over time following disturbance [13,43,45]. Within ground-dwelling arthropods, evidence indicates that dominant assembly processes can vary among different assemblages of the larger community [13], as well as across species with different ecological strategies, relative abundances, and dispersal rates [17,18,39,41,42,46–48].

In mixed-conifer forests of the southwestern United States (U.S.), historical disturbance regimes would have included frequent fires [49–52]. For example, tree-ring data has indicated a fire return interval ranging from four to 26 years across forests of Arizona and New Mexico, United States (USA) [53], and from three to 14 years in the more southerly forests of New Mexico [54]. Prior to widespread fire suppression, these historical fire regimes would have created forest mosaics of different successional states and a relatively high proportion of recently burned areas [55]. More recently, fire suppression has greatly lengthened fire return intervals with high-severity fires, especially repeated burns over relatively short periods, which have promoted shifts from pine-dominated and mixed-conifer forests toward communities of shrubs and grasses across the southwest [49–52]. Studies of wildfire effects on ground-dwelling arthropods in this region remain rare, but evidence indicates that changes in litter and fuel characteristics due to wildfires can impact ground-dwelling arthropod communities in pinyon–juniper woodlands [56]. Yet how wildfires affect arthropod communities in other southwest forest types remains unclear.

Given that reference conditions encompassing fire regimes with historically natural severity and return intervals are not available in this region, and that comparing samples from within the boundaries of large wildfires to unburned areas with different pre-fire conditions is likely to introduce variation, we chose to explore the assembly of post-wildfire, ground-dwelling arthropod communities in areas affected by either one or two recent wildfires. Specifically, we assessed the influence of post-fire understory characteristics on arthropod abundance, taxon diversity, community structure, and the processes governing the assembly of arthropod communities from samples collected in sites of northern New Mexico's Jemez Mountains burned by the 66.8 km$^2$ Dome Fire in 1996, by the 190 km$^2$ Cerro Grande Fire in 2000, and in areas of these prior wildfires that were burned again in 2011 by the 630 km$^2$ Las Conchas Fire. Previous work in conifer forests has revealed an increased influence of stochastic

assembly processes in ground-dwelling communities shortly after disturbance, with a shift toward a stronger role of deterministic processes over time [13,45]. Thus, we predicted that ground-dwelling arthropod communities within areas burned by the Las Conchas Fire, five years prior to sampling, would be characterized by a greater relative influence of stochastic processes compared to communities from areas burned ≥16 years prior to our study by the Dome and Cerro Grande Fires.

## 2. Materials and Methods

### 2.1. Site Description

Our study was conducted in 2016 between 2250–2750 m above sea level in the Jemez Mountains, New Mexico, USA. This area is semi-arid, and a majority of precipitation occurs as warm-season monsoons. Pre-settlement fire regimes included a substantial component of low-severity fire with short return intervals [57]. We considered the influence of recent wildfires on arthropods by sampling in four "fire groups" representing areas burned by one or two recent wildfires: (1) Dome Fire (DM), (2) Cerro Grande Fire (CG), (3) Dome Fire + Las Conchas Fire (DMLC), and (4) Cerro Grande Fire + Las Conchas Fire (CGLC).

### 2.2. Experimental Design and Arthropod Sampling

To assess arthropod responses, we established 20 transects, each 50 m in length, with four located in the CG fire group, six in the CGLC fire group, and five each in the DM and DMLC fire groups. Ground-dwelling arthropods were sampled using five pitfall traps placed at the 5-m, 15-m, 25-m, 35-m, and 45-m transect locations. Traps were 80 mm in diameter and 120 mm deep, and consisted of a funnel inserted into a cup containing a 100:200 mL mixture of propylene glycol and $H_2O$ as a preservative, and were installed with their openings just below the soil surface. To reduce litter and small-vertebrates capture, traps were covered with $200 \times 200$ mm wood covers, with a 20-mm opening between the cover and ground surface. Traps were opened for two, 28-day periods divided among the early-summer (dry) and late-summer (monsoon) seasons. Traps were emptied and preservative replenished on day 14 and emptied and removed on day 28 of each period. Prior to analysis, the two 28-day intervals were binned into one aggregate sample for each trap. Most of the arthropods were identified to their family with the exception of Formicidae (identified to genus), Orthoptera (identified to suborders with Caelifera identified to family), and Archaeognatha, Araneae, and Opiliones (identified to order). Several taxonomic groups were excluded from analyses (Table S1) given their potential attraction to preservative liquid (e.g., bees) or volatiles from decomposition (e.g., carrion beetles), or because they are not well-sampled with pitfall traps (e.g., Acari) [58,59].

### 2.3. Understory Environmental Assessment

Environmental and vegetation characteristics were assessed along five-meter transects extending in four cardinal directions from each trap. Measures included the cover and quantities of duff and litter, tallies of fine and coarse woody surface fuels, counts and area of cover by tree trunks, and ocular estimates of vegetation cover by forbs, grasses, and shrubs. Burn severity, which was measured as the change in the relativized burn ratio (RBR), was assessed from remote sensing imagery [51]. When crossing one of the five-meter transect lines, woody fuels—i.e., coarse woody debris (CWD)—were assigned to time-lag categories of 1-h, 10-h, 100-h, and 1000-h fuels following the protocols of Brown [60], with these categories representing the time needed for a given piece of CWD to equilibrate with ambient relative humidity, assuming static weather conditions. Occurrences of individual species of trees and woody shrubs crossing the transect lines were also recorded.

### 2.4. Data Analysis

All analyses were performed in R version 3.4.3 ("Kite-Eating Tree") [61]. We used the "vegan" package [62] to generate a taxon accumulation curve, calculate the Shannon diversity and sample

dissimilarities, and compare arthropod communities among fire groups via PERMANOVA on Bray–Curtis distance matrices with traps nested in their respective transects to account for spatial variation. We also used "vegan" to compute null models via the "oecosimu" function with Bray–Curtis dissimilarities, and the "c0_samp" non-sequential algorithm that keeps column sums constant, while cells within each column are shuffled. Using mean dissimilarities measured within the null communities after 999 simulations, we calculated a null deviation value for each fire group as (βobs − βnull)/βnull [45].

We measured the influences of transect, environmental factors, and vegetation characteristics on community structure using stepwise (variables improving the model $R^2 \geq 0.02$ were retained) fuzzy set ordination (FSO) [63] in the package "fso" for R [64]. We completed a metacommunity analysis in the "metacom" package [65], where "coherence" was calculated from the number of embedded absences in the ordinated empirical incidence matrix to a distribution of embedded absences derived from 1000 ordinated null matrices, "turnover" was measured by counting the number of taxon replacements of one species by another with significance calculated via comparison to values from 1000 null models of replacement, and boundary clumping was measured using Morisita's index, in which a measure of the dispersion of species occurrences among sites [65] with significance determined via a chi-square test comparing the observed distribution to a uniform distribution.

Indicator species analysis was performed with the "indicspecies" package [66] to determine which taxa displayed significant relationships to each fire group. We used the "lme4" package in R [67] to construct linear mixed-models for (1) arthropod total abundance (log transformed to improve the distribution prior to modeling), (2) Shannon diversity (H´), and (3) fire severity effects on both logs and duff quantity (both of which were log transformed to improve their distribution prior to modeling). In these models, we considered the response of each variable as a function of fire group (fixed effect) and sampling transect (random effect) as a basic model that was compared via corrected Akaike information criterion (AICc) values to more complex models, which included additional fixed effects identified by random forest classification as explaining ≥0.02 of variation (Table S2). Random forests were performed in the "randomForest" package [68] with predictors ranked using the "caret" package [69]. The complex mixed-model was stepwise reduced using the "lmerTest" package [70] to identify a best-fit model that was then compared to the null model via AICc value. Conditional and marginal $R^2$ values for each model were computed using the "MuMIn" package [71]. Since metrics of beta diversity rely on pairwise comparisons among samples, the relationship of arthropod sample dissimilarity to understory characteristics was analyzed via a permutational linear model in the "lmPerm" package [72] to avoid violating the common general linear model assumption of sample independence. We also used "lmPerm" for comparisons of null deviation values among fire groups, and the "DAAG" package [73] to determine if the null deviation distributions within fire groups differed from zero. Finally, to assess the degree to which fire groups could be differentiated by environmental and vegetation characteristics, we completed quadratic discriminant analysis in the "MASS" package [74].

## 3. Results and Discussion

A total of 26,383 individual arthropods were captured (following the removal of non-target groups). Sampling was considered adequate based on the taxonomic accumulation curve (Figure S1). In total, there were 81 unique taxon groups identified across all the samples. Subtotals of 56, 62, 57, and 59 of these taxa were found in the CG, CGLC, DM, and DMLC fire groups, respectively. We found that the multivariate community structure differed significantly among all the fire groups based on PERMANOVA ($p < 0.001$ for all pairwise comparisons). In addition, we also found a significant effect of sampling transects on community structure ($p < 0.001$). The relationship of community structure to environmental variables was examined using stepwise fuzzy set ordination (FSO), which also identified transect as a significant factor along with several additional variables ($R^2 = 0.50$, $p < 0.001$ for the full

model). Factors retained in the final FSO model included—in order of importance—graminoid cover, quantity of duff (mg/ha), quantity of litter (mg/ha), tree basal area, and 1000-h fuels (mg/ha) (Figure S2).

Arthropod abundance did not differ significantly among the fire groups or in relation to the measured understory properties, as the intercept-only (null) model containing transect as a random effect was considered to be the best-fit based on AICc values within a stepwise fitting procedure. For arthropod Shannon diversity, duff quantity (mg/ha) ($F = 4.23$, $p = 0.042$) was the only fixed-effect retained in the best-fit model, which had a marginal $R^2 = 0.05$ (variance explained by the fixed effects) and a conditional $R^2 = 0.53$ (variance explained by the full model with transect as a random effect). Arthropod dissimilarity among samples—i.e., the turnover in taxon among sample locations measured as a mean pairwise Bray–Curtis distance—had a best-fit permutational model with an adjusted $R^2 = 0.43$ ($p < 0.0001$) and retained three significant factors ($p < 0.05$), including fire group (Figure S3), rock cover, and the number of logs on the ground (Table 1). In the case of taxon diversity, increasing quantities of duff had a positive effect on Shannon diversity, while arthropod dissimilarity had a positive relationship with both logs and rock cover (Figure 1).

**Table 1.** Best-fit permutational model relating Shannon diversity responses to fire groups and understory characteristics.

| Factor | DF | SS | Iterations | *p*-Value |
|---|---|---|---|---|
| Fire group | 3 | 0.113 | 5000 | <0.0001 |
| Logs (count) | 1 | 0.036 | 5000 | <0.0001 |
| Rock cover | 1 | 0.034 | 5000 | <0.0001 |
| Residuals | 94 | 0.204 | | |

Values are from a permutational linear model fit with "lmp()" in the "lmPerm" package for R.

Given the response of arthropod diversity to duff quantity and of arthropod dissimilarity to the number of logs and rock cover, we considered possible influences of fire severity (with sample transect as a random effect) on these understory properties. While fire severity did not have a significant effect on exposed rock cover, fire severity did have a significant, negative effect on duff quantity ($p = 0.012$) and a significant, positive effect on the number of logs on the ground ($p = 0.018$). For duff, the mixed model had a marginal $R^2 = 0.22$ and a conditional $R^2 = 0.72$, while for logs, the marginal $R^2 = 0.17$ and conditional $R^2 = 0.61$. Collectively, these models indicate an important influence of burn severity alongside the spatial influences of transect locations on these surface fuel measures. While modest in effect, the positive relationship between duff quantity and taxon diversity, as well as that between logs and taxon dissimilarity suggests that increasing burn severity could lead to a less diverse ground-dwelling arthropod community with greater rates of taxon turnover among locations, at least in part due to the burn severity impacts on surface fuels—which is an outcome pointing to an important influence of the spatial heterogeneity of fire effects on biodiversity in this system. However, these results also highlight the challenge for teasing apart the potential influences of pre-existing environmental heterogeneity from wildfire effects on understory biotic communities. Thus, it is important to consider that the important influence of sampling transects revealed by the mixed-models for duff quantity and log counts likely reflect the interaction of pre-fire conditions and wildfire behavior that collectively determined environmental heterogeneity across the study landscape, which is a result that has been previously reported for patterns in post-fire vegetation responses in our study area [51].

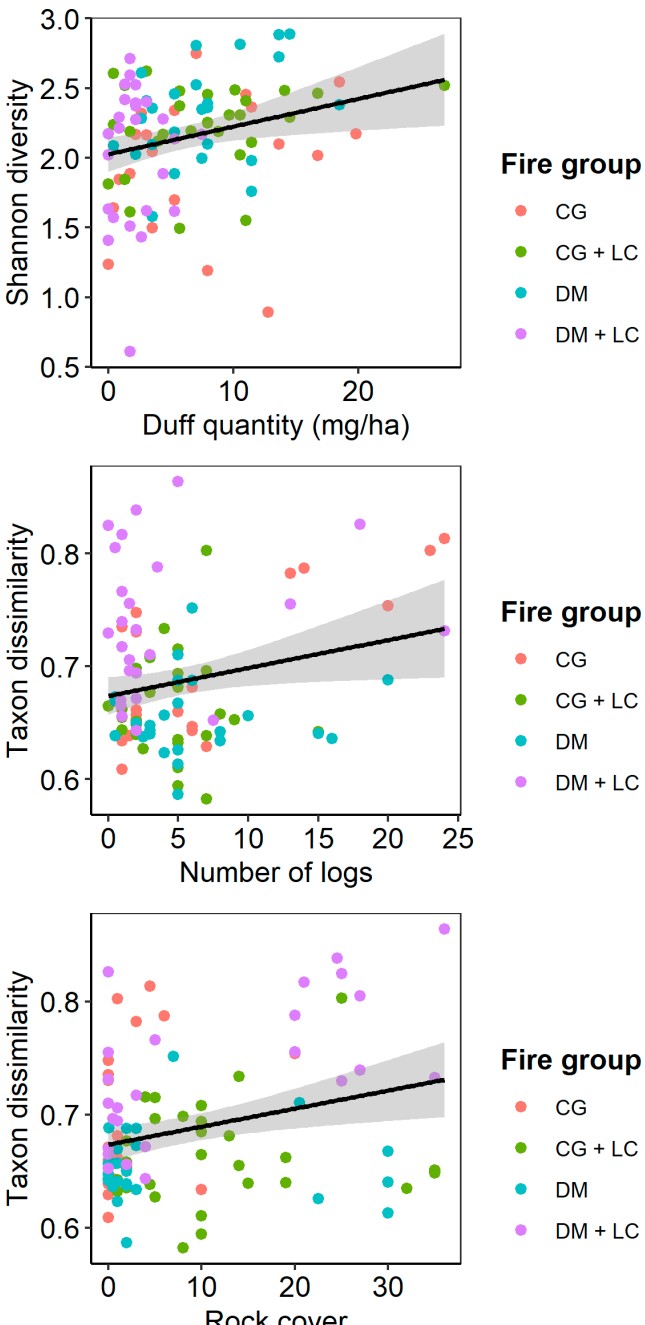

**Figure 1.** Relationships of taxon diversity to duff quantity (mg/ha) (top-panel) and of taxon dissimilarity among arthropod samples to the number of logs and rock cover (middle and bottom panels, respectively) at sample locations across fire groups. Symbol colors indicate the different fire groups, and shading around the line of fit indicates the 95% confidence intervals.

Understory environmental and vegetation factors have been shown to affect ground-dwelling arthropods in both unburned [75–78] and burned landscapes [12,79]. However, post-fire understory characteristics have also been reported to exert a minimal influence on ground-dwelling arthropod communities in some locations [80]. Despite explaining a maximum of 50% of variation in the ground-dwelling arthropod community structure (a result of the FSO) and lesser amounts of Shannon diversity and taxon dissimilarity (Figure 1), understory characteristics were important factors for differentiating the fire groups (Figure 2). A best-fit quadratic discriminant analysis model (QDA) correctly classified 84% of trapping sites to fire group based on a set of seven variables: cover of rocks and forbs, number of dead trees, and quantities of duff, litter, and 1000-h and 10-h fuels. By fire group, the QDA correctly classified 85% of CG, 77% of CG + LC, and 88% of both DM and DM + LC.

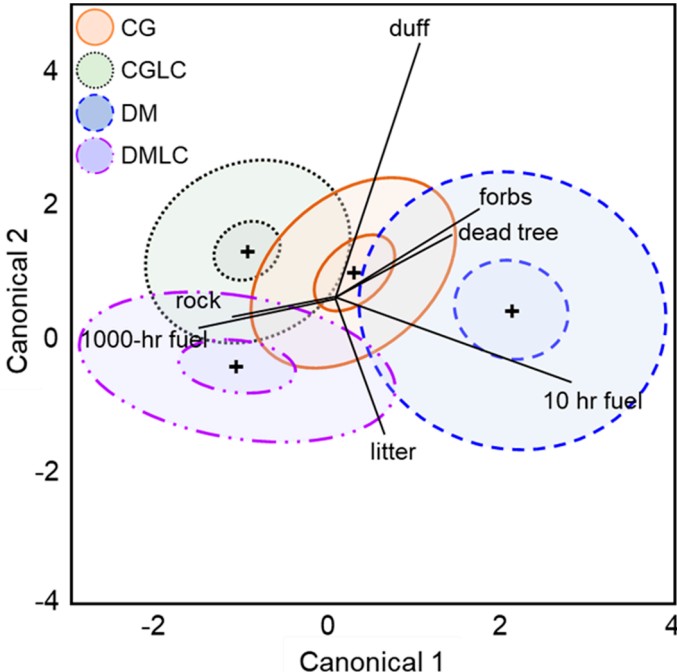

**Figure 2.** Canonical plots illustrating quadratic discriminant analyses (QDA) of multivariate measures of habitat/environment of sampling locations in the Cerro Grande (red ellipses), Dome (blue ellipses), and the Cerro Grande + Las Conchas fires (green ellipses) and Dome + Las Conchas fires (orange ellipses). QDA successfully categorized 84% of samples into the correct fire group. The multivariate means of each group are denoted by plus signs (+), smaller ellipses around each mean illustrate the canonical space containing 50% of the group observations, and larger ellipses illustrate 95% confidence levels. The labeled rays, scaled to 200% of their actual length, show the directions in the canonical space of the seven covariates retained in the best-fit QDA model.

While only a portion of variation in the full community and diversity of ground-dwelling arthropods were explained by post-fire understory characteristics, the potential for specific taxonomic groups to respond to the understory environment following fires remains likely. Using indicator species analysis, we found significant association among four taxa in the CG group, seven in the CGLC group, eight in the DM group, and four in the DMLC fire group leading to a range of 6.4% to 14.0% of taxa considered to be significant indicators of their respective fire group (Table 2).

**Table 2.** Potential indicator taxa of different fire groups based on indicator values and permutational tests of significance.

| Fire group | Order | Suborder/Family | Spec. | Prob. | Stat | *p* |
|---|---|---|---|---|---|---|
| Cerro Grande (CG) | Coleoptera | Carabidae | 0.66 | 0.90 | 0.77 | 0.005 |
| | Coleoptera | Unknown larvae | 0.55 | 0.65 | 0.60 | 0.035 |
| | Hemiptera | Nabidae | 0.65 | 0.40 | 0.51 | 0.005 |
| | Hemiptera | Miridae | 1.00 | 0.15 | 0.39 | 0.010 |
| Cerro Grande + Las Conchas (CG + LC) | Hymenoptera | Tapinoma | 0.53 | 0.87 | 0.68 | 0.005 |
| | Orthoptera | Ensifera | 0.51 | 0.83 | 0.65 | 0.005 |
| | Orthoptera | Rhaphidophoridae | 0.44 | 0.97 | 0.65 | 0.010 |
| | Araneae | Gnaphosidae | 0.39 | 0.93 | 0.60 | 0.025 |
| | Scolopendromorpha | | 0.40 | 0.67 | 0.52 | 0.045 |
| | Coleoptera | Scarabidae | 0.60 | 0.37 | 0.47 | 0.015 |
| | Coleoptera | Pselaphinae | 1.00 | 0.13 | 0.37 | 0.025 |
| Dome (DM) | Opiliones | | 0.55 | 0.88 | 0.69 | 0.005 |
| | Hemiptera | Cicadellidae | 0.50 | 0.76 | 0.61 | 0.015 |
| | Hemiptera | Aphidae | 0.54 | 0.60 | 0.57 | 0.005 |
| | Coleoptera | Byrrhidae | 0.74 | 0.40 | 0.55 | 0.005 |
| | Coleoptera | Curculionidae | 0.44 | 0.64 | 0.53 | 0.040 |
| | Hymenoptera | Myrmicinae | 0.96 | 0.24 | 0.48 | 0.010 |
| | Coleoptera | Cryptophagidae | 0.61 | 0.36 | 0.47 | 0.005 |
| | Coleoptera | Ptilidae | 0.75 | 0.28 | 0.46 | 0.030 |
| Dome + Las Conchas (DM + LC) | Hymenoptera | Pheidole | 0.75 | 0.68 | 0.72 | 0.005 |
| | Coleoptera | Anthicidae | 0.73 | 0.52 | 0.62 | 0.005 |
| | Archaeognatha | | 0.70 | 0.40 | 0.53 | 0.005 |
| | Hymenoptera | Solenopsis | 0.42 | 0.64 | 0.52 | 0.040 |

Spec. = "Specificity" and indicates the probability that samples containing the listed taxa belong to the respective fire group; Prob. = "Probability" and is the probability of finding the given taxa in a sample within the respective fire group.

Response models for each of the significant indicator taxa revealed that abundance patterns for five of the 23 indicator taxa were significantly related to understory characteristics (full models having a *p* < 0.05) based on permutational regression models—with each model containing soil and rock cover, all of the CWD categories, litter and duff measures, dead and live tree occurrence, and cover by plant functional groups. Within the CG fire group, both of the beetle taxa were significantly related to post-fire understory characteristics with the Carabidae and unknown larvae group having an $R^2$ = 0.97 and 0.95 respectively. For the CG + LC fire group, models suggested no significant relationships of any indicator taxa to the understory characteristics considered. Within the DM fire group, both the Opiliones and the Myrmicinae ant group were significantly related to the understory characteristics with $R^2$ = 0.81 and 0.62, respectively. Finally, in the DM + LC fire group, Solenopsis ants were significantly related to the understory characteristics with an $R^2$ = 0.93. The relationship of Carabid beetles and two genera of Formicidae to the understory characteristics tested joins a number of reports of the post-fire environmental characteristics affecting these groups [28,56,80–83].

Previous work in mixed-conifer forests subjected to prescribe fire [80] and piñon-juniper woodlands affected by wildfire [56] indicates that burning tends to increase the proportion of indicator arthropod taxa. We found similar overall ranges in indicator values as those in these studies, but found a smaller proportion of taxa to be significant indicators following wildfire compared to Higgins et al. [56]. Possible explanations for this difference are: (a) some of our sampling was completed in sites with more time to recover since fire; (b) we lumped our arthropods into higher taxonomic groupings than Higgins et al., thereby obscuring some species-level information regarding habitat preference; (c) our study covers gradients of fire severity and areas with high pre-fire environmental heterogeneity within each fire group that collectively introduced variation and led to a greater overlap in the post-fire environmental conditions among groups; and (d) stochastic community assembly processes are important drivers

across this landscape, leading to a reduced role of environmental filtering and niche-matching in determining the ground-dwelling arthropod community structure.

Despite the potential for stochastic assembly processes to shape arthropod communities particularly after disturbances [13], the dispersal and probability of colonization by non-pest arthropods at the landscape-scale in forest ecosystems remains poorly studied [13,84,85]. If the stochastic processes related to the probability of local extinction or to the probability of dispersal and recolonization from source populations following disturbances strongly influence the taxa in this system, a dampening in the niche-based associations in these communities should be noted following fire. Nevertheless, an apparent lack of strong relationships among understory characteristics and the majority of indicator taxa and taxon diversity does not eliminate the possibility that unmeasured aspects of the understory environment could be drivers of arthropod abundance and distribution in these burned landscapes. However, if unmeasured environmental characteristics are important drivers of community assembly, coherent boundaries and predictable gradients in taxon turnover should be present in a metacommunity analysis.

Based on a null modeling approach that compares observed levels of taxon turnover (dissimilarity among samples) to turnover within a randomized community matrix via a null model, we found an indication of a stronger relative influence of stochastic processes on arthropod community assembly in the CG + LC fire group than in the CG, DM, and DM + LC groups, which significantly differed from the null model (Figure 3). While this result supports the role of both deterministic and stochastic processes in the assembly of ground-dwelling arthropods following wildfires, it also suggests that neither arthropod responses to multiple wildfires, nor the time since the most recent wildfire are generalizable drivers of community assembly in this system. While null deviation values with a greater departure from zero invoke a stronger relative influence of deterministic processes in community assembly, values can also indicate communities that are either more similar than expected by chance (negative values), or less similar (positive values) [13,40,45]. Thus, our results suggest that arthropod diversity and community structure in the CG and DM + LC fire groups are potentially responding to unmeasured aspects of the understory environment that are promoting assemblages of similar taxa from among the larger arthropod community of the study area. Alternatively, in the DM fire group, the positive null deviation values invoke processes, such as interspecific competition for resources, that lead to less similar taxa within samples than those expected by chance. However, it is important to note that while the distribution of null deviation values for the DM fire group significantly differ from zero, the group's values still overlap zero near the first quartile (Figure 3), suggesting a stronger relative influence of stochastic processes in the assembly of the arthropod community as a whole compared to the CG and DM + LC fire groups, which significantly differ from the DM fire group [13,45].

Similar to the null deviation analysis, the metacommunity analysis provided further evidence that the arthropod communities of the CG + LC and DM fire groups have relatively random patterns of species presence and absence based on the non-significant values of coherence, which, as a measure, indicates that most taxa are not cohesively responding to an environmental gradient (Table 3) [86]. However, within the CG and DM + LC fire groups, the significant, positive values of coherence indicate that a majority of taxa are sorting along an environmental gradient, with the significant negative values of metacommunity turnover indicating a high degree of change in the taxa present at each end of this environmental gradient. This outcome dismisses the presence of traditional Clementsian-type or Gleasonian-type niche matching by the arthropod taxa in these groups [86]. Finally, based on the assessment of boundary clumping, taxa within the CG and DM + LC fire groups appear to be "clumped" together into assemblages within the larger community. This result agrees with the primarily negative values of null deviation found for these groups, which indicated arthropod communities that were more similar than chance at the sample level (Figure 3). Combining all three of the metacommunity metrics and the null deviation analysis suggests that taxa from less diverse sites within these fire groups are nested subsets of taxa from more diverse sites [86]. This result supports a stronger role of species sorting/deterministic factors at smaller scales and a stronger role of mass effects linked to a

decreasing probability of dispersal into and the colonization of suitable habitats as these resources patches become more isolated at larger spatial scales [87].

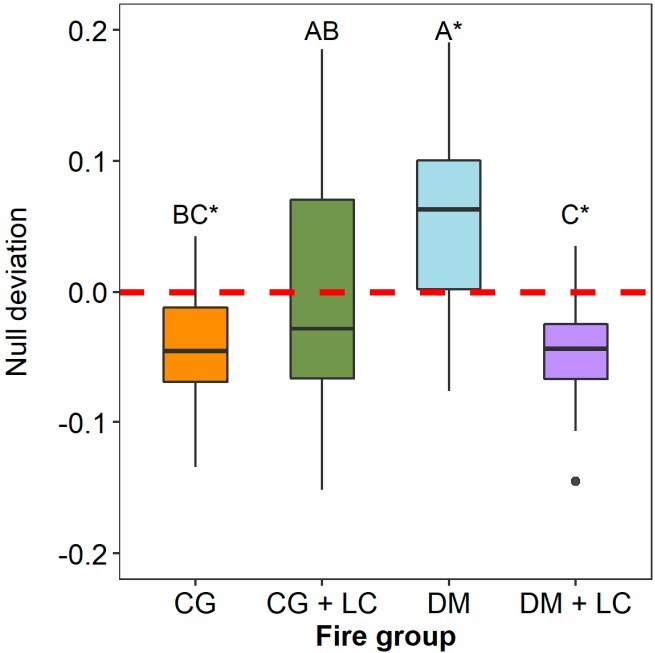

**Figure 3.** Null deviation values for the once-burned and twice-burned wildfire groups, including the Cerro Grande fire (CG), the Dome fire (DM), the Cerro Grande + Las Conchas fires (CG + LC), and the Dome + Las Conchas fires (DM + LC). Increasing null deviation values, positive or negative, indicate larger influences of deterministic community assembly processes such as environmental filtering or competition/facilitation. A value of zero indicates no difference between observed and null/randomized taxon dissimilarities; thus, values closer to zero indicate the increasing relative importance of stochastic assembly processes. Lettering indicates statistical comparisons among fire groups via a permutational post hoc test; an asterisk following letters indicates that the group's distribution of null deviation values significantly differs from zero via a permutational one-sample *t*-test.

**Table 3.** Measures of species coherence, turnover, and boundary clumping in ground-dwelling arthropod metacommunities sampled in sites burned by either one or two (reburned) wildfires.

| | Coherence | | | | | Turnover | | | | | Bound. Clumping | | |
|---|---|---|---|---|---|---|---|---|---|---|---|---|---|
| **Fire** | **Abs.** | **Z** | **p** | **x̄** | **SD** | **Rep.** | **Z** | **p** | **x̄** | **SD** | **Mor. Ind.** | **p** | **Pattern** |
| CG | 365 | 4.0 | <0.01 | 484.5 | 5.5 | 11,935 | −4.6 | <0.01 | 6558.0 | 34.0 | 1.77 | <0.01 | Nested |
| CG + LC | 806 | 0.1 | 0.96 | 808.4 | 6.8 | | | | | | | | Random |
| DM | 521 | 1.7 | 0.08 | 580.0 | 5.8 | | | | | | | | Random |
| DM + LC | 532 | 3.5 | <0.01 | 660.7 | 6.1 | 17,285 | −4.0 | <0.01 | 10312.7 | 41.8 | 1.69 | <0.01 | Nested |

CG = Cerro Grande fire, DM = Dome fire, and CG + LC and DM + LC = Areas where the Las Conchas fire burned within the footprint of either the Cerro Grande or Dome fires; Abs. = embedded absences of species, Rep. = number of species replacements, Mor. ind. = Morisita's index.

## 4. Conclusions

While deterministic processes are of clear importance in structuring biotic communities and commonly a first consideration among ecologists [32], stochastic assembly processes have long been known to shape community composition and diversity patterns [33–37]. Disturbances have been invoked as a primary moderator of assembly processes [40,43–45]. Indeed, recent work from the conifer forests of Colorado, USA, has demonstrated an amplified importance of stochastic processes in the assembly of ground-dwelling arthropod communities following tree mortality during bark

beetle epidemics [13] and in soil microbial communities following wildfires [45], with the balance of deterministic and stochastic influences shifting over time since the disturbances in both of these studies. Also, time since the wildfire was an important driver of ground-dwelling arthropod abundance and diversity in the piñon-juniper woodlands of northern New Mexico, USA [56]. While we found that ground-dwelling arthropods in areas burned by one or two recent wildfires exhibited patterns of both stochastic and deterministic processes, we did not find support for a generalizable result across fire groups based on the number of recent wildfires or time since disturbance. Overall, we found a moderate to weak influence of understory abiotic and vegetation factors on arthropod community structure. However, we cannot dismiss a potential influence of unmeasured aspects of the understory environment. For example, lagged effects from the spatiotemporal turnover in plant species [12] or the microclimatic variation that affects arthropod physiology and alters species interactions at local to regional scales [88] are among the possible unaccounted-for factors governing community patterns in this system. Also, work in managed deciduous forests indicates that different types of diversity can peak at alternate ends of environmental gradients—e.g., spider compositional diversity increased as oak forest canopy became sparser, while spider functional diversity increased under closed canopy [89]. Thus, there remains the potential that a stronger relative influence of deterministic processes may become apparent if community structures were to be assessed via functional traits instead of via taxonomy. At the same time, disentangling the potential influences of pre-existing environmental heterogeneity from wildfire effects on understory biotic communities was not possible given our study design. As a result, the significant influence of sampling location on community structure and diversity that we observed could reflect an influence of pre-fire arthropod taxa diversity and turnover across the study landscape on post-fire community patterns. However, the apparent importance of stochastic processes in arthropod community assembly in two of the four fire groups we studied highlights the potential for dispersal rates from refugia [90], random or coarse spatial patterning of source populations [91,92], or a reduction in the importance of biotic interactions as filters following disturbance [13] to shape ground-dwelling arthropod communities in this system. Given the indications that both deterministic and stochastic processes were of variable importance across areas impacted by similar disturbances, additional effort to identify the processes governing diversity and composition following disturbance is necessary for understanding the factors controlling the ground-dwelling arthropod community structure in this wildfire-prone landscape.

**Supplementary Materials:** The following are available online at http://www.mdpi.com/1999-4907/10/8/667/s1, Figure S1: Taxon accumulation curve showing unique groups captured as a function of samples collected, Figure S2: Fit of ordination distance from fuzzy set ordination (FSO) to dissimilarity in the original distance matrix of the ground-dwelling arthropod community, Figure S3: Unimetric measures of arthrpod abundance, Shannon diversity, and mean pairwise dissimilarity (a metric of beta-diversity measured as Bray-Curtis distance) among the four fire groups, Table S1: List of captured taxa excluded from analyses, Table S2: Random forest results for predictions of arthropod abundance, Shannon diversity, and taxon dissimilarity.

**Author Contributions:** Conceptualization, J.D.C. and P.W.; Methodology, P.W. and J.D.C.; Data Analysis and Visualization, S.F.; Data Curation, P.W. and S.F.; Writing—Original Draft Preparation, S.F.; Writing—Review & Editing, S.F., P.W. and J.D.C.; Funding Acquisition, J.D.C.

**Funding:** This research was funded by the Western Colorado University's Master in Environmental Management (MEM) Haley Fund and an agreement between the U.S. Forest Service Rocky Mountain Research Station, Aldo Leopold Wilderness Research Institute and Western Colorado University (15-CR-11221639-118).

**Acknowledgments:** We thank the many individuals who helped with data collection and arthropod identifications. We thank the three anonymous reviewers for comments that improved our manuscript.

**Conflicts of Interest:** The authors declare no conflict of interest. The sponsors had no role in the design, execution, interpretation, or writing of the study.

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
