# Peer review of "Ground-Dwelling Arthropod Community Responses to Recent and Repeated Wildfires in Conifer Forests of Northern New Mexico, USA"

_forests, doi:10.3390/f10080667_

Round 1

Reviewer 1 Report

This is nice study with clear results which add to current inventory studies concerning impact of wilfires on arthropods. Until now, no study focused on arthropod communities in the studied habitats. Statistical analyses are robust and described in very details, also results are very well presented. I do not have any criticism and I support acceptation of this MS.

I only have minor points:

I am missing better descrtiption of studied arthropod communities (which groups and their importance for indication in habitats under fire influence).

(Line 66-70) Add references concerning arthropod inhabitation of  habitats under various disturbances including active management and fire - reporting short-term effect: Samu et al., 1999 (https://www.jstor.org/stable/3706004?seq=1#page_scan_tab_contents); Košulič et al., 2016 (https://journals.plos.org/plosone/article?id=10.1371/journal.pone.0148585#references). 

Author Response

This is a nice study with clear results which add to current inventory studies concerning the impact of wildfires on arthropods. Until now, no study focused on arthropod communities in the studied habitats. Statistical analyses are robust and described in very details, also results are very well presented. I do not have any criticism and I support acceptation of this MS.

We appreciate the compliment.

I only have minor points:

I am missing better description of studied arthropod communities (which groups and their importance for indication in habitats under fire influence).

In the paragraph beginning on line 258, we mention some common indicator taxa groups—we did not include species-level information here as this is expected to vary greatly among habitats as a reflection of biogeography patterns.

(Line 66-70) Add references concerning arthropod inhabitation of habitats under various disturbances including active management and fire - reporting short-term effect: Samu et al., 1999 (https://www.jstor.org/stable/3706004?seq=1#page_scan_tab_contents); Košulič et al., 2016 (https://journals.plos.org/plosone/article?id=10.1371/journal.pone.0148585#references).

We appreciate having these papers brought to our attention. 

We have not included either in the section where line numbers are noted because this section is specifically about studies that have revealed important roles of stochastic processes in arthropod community assembly.  

We have included the Košulič et al., 2016 reference on lines 370 - 374 as it shows that compositional diversity and functional diversity can respond to different aspects of environmental gradients which is an interesting finding and an important point for readers to consider in this section. This was a great paper with a very interesting set of results and one we sadly missed in our earlier search of the literature.

We are opting to forgo adding Samu et al. 1999 because it describes arthropod abundances in relation to borders and vegetation strips in agricultural settings and while this is a strong paper and was a fun read, we did not feel that it was directly transferable to forest ecosystems. Given the large literature on arthropods in agricultural settings, we wish to avoid appearing to highlight a specific study or concept without being able to summarize this larger literature that we feel would fall beyond the scope of our study about fire impacts on arthropods in conifer forests and open us to criticism by readers who know the agricultural literature far better than we do.

Reviewer 2 Report

The authors have conducted a significant study examining arthropod distributions in conifer forests of northern New Mexico. Overall, I think the manuscript is well written, but I request that the authors address some key issues before publication. Here are my main points for reflection:

First, on lines 87 and 88, you state that you “chose to explore the effects of wildfire on ground-dwelling arthropod community assembly”. I think that this should be reworded, as you did not examine the “effects of wildfire”. You instead, are exploring community assembly following wildfire. I think this is a subtle, but important point, as you would need areas as controls to be able to say anything significant about the influence of fire. This is connected with my next key issue I would like to see you address.

Second, I am having a hard time deciphering how much of the difference in community structure is associated with variables influenced by the fire and how much of the difference in community structure was associated with differences in arthropod community structure that existed across the landscape prior to the fire. I think this needs to be explicitly addressed in your discussion. In addition, to stocasiticity in dispersal and survivorship, there could easily have just been different assemblages at your sites prior to the fire. Your data cannot tease these two things apart, as such, you need to really spell this out as a possibility. This idea is also connected to your indicator analyses. Do indicator species describe different fire regimes, differences in current habitat characteristics, or are they indicative of more historical habitat characteristics? Areas with more exposed rock would have species that utilize this habitat before and after the fire. Similarly, areas with lots of wood following the fire, probably had more dense forests before the fire. I know you did address some of these concerns in your manuscript, but I think they really need to be more of a focal point in the discussion.

I may be missing something, but I would prefer that you reexamine your interpretation of Figure 3 on lines 284-286 and your statistical approach. The key here is if the different fire groups differ from the null hypothesis, not each other. Do any of these significantly differ from zero? If not, then it seems that stochastic processes are the main driver for each of the four fire groups. The differences among fire groups is really not important at that point. This should influence your conclusions.

The patterns presented in figure 1 are interesting at least from the “eye ball test”. First, I think that including fire severity should be removed. It just complicates the figures. Second, I think it would be interesting to test these patterns for each fire group. I think this might provide a more interesting examination of what is happening. Take for instance rock cover for DM+LC, there certainly is a positive relationship, but I do not think there is for the other burn categories. Is there something different about DM+LC? Similarly, for No. of Logs, taxon dissimilarity seems positively correlated for CG, but not for the other sites? Is there something about CG, or is this an important factor influencing arthropods across all your sites?

Finally, the conclusion needs to be re-thought. The statement on lines 340-344 is of course possible, but if these variables are known to be key, why were they not measured? Rethink if stocasiticity is influencing arthropods in two and not all sites. Also, any statements on biological filters are just conjecture.  In general, I think that the interesting aspects of this paper are that there are few predictors for arthropod community structure following disturbances. It seems that stochastic processes are probably the best model for arthropod community structure following a disturbance (although differences in arthropod community structure prior to the disturbance may also be important but unexplored in this study). Overall, your results highlight the complexities in predicting and understanding arthropod diversity across a disturbed landscape.

Author Response

First, on lines 87 and 88, you state that you “chose to explore the effects of wildfire on ground-dwelling arthropod community assembly”. I think that this should be reworded, as you did not examine the “effects of wildfire”. You instead, are exploring community assembly following wildfire. I think this is a subtle, but important point, as you would need areas as controls to be able to say anything significant about the influence of fire. This is connected with my next key issue I would like to see you address.

We agree that this is a more accurate description of our study and we’ve made this change (line 88) and made a reference to arthropod responses, not wildfire effects in relevant points throughout.

Second, I am having a hard time deciphering how much of the difference in community structure is associated with variables influenced by the fire and how much of the difference in community structure was associated with differences in arthropod community structure that existed across the landscape prior to the fire. I think this needs to be explicitly addressed in your discussion. In addition, to stochasticity in dispersal and survivorship, there could easily have just been different assemblages at your sites prior to the fire. Your data cannot tease these two things apart, as such, you need to really spell this out as a possibility. This idea is also connected to your indicator analyses. Do indicator species describe different fire regimes, differences in current habitat characteristics, or are they indicative of more historical habitat characteristics? Areas with more exposed rock would have species that utilize this habitat before and after the fire. Similarly, areas with lots of wood following the fire, probably had more dense forests before the fire. I know you did address some of these concerns in your manuscript, but I think they really need to be more of a focal point in the discussion.

This is an important point and one we thought that had been made clear, but realize, as you have noted, was not explicit enough. We have addressed this topic more directly in the conclusions section, lines 381 – 386, where we think readers are more likely to focus their attention and therefore more likely to see this caveat addressed. However, we also make note of this topic on lines 223 - 229 in part as a  response to your upcoming comment on the patterns in Figure 1. We see these comments and the necessary responses as being somewhat related as they both highlight the possible interactions of pre-existing and post-fire heterogeneity in biotic and abiotic factors.

I may be missing something, but I would prefer that you reexamine your interpretation of Figure 3 on lines 284-286 and your statistical approach. The key here is if the different fire groups differ from the null hypothesis, not each other. Do any of these significantly differ from zero? If not, then it seems that stochastic processes are the main driver for each of the four fire groups. The differences among fire groups is really not important at that point. This should influence your conclusions.

We agree this is useful information and overlooked including such a test/indicator. We have added an “*” following the among group lettering to indicate groups with null deviation distributions that significantly differ from zero. This new test was performed with a permutational one-sample t-test in the R package “DAAG”. It remains important for readers to understand that the values in this figure are intended to be an index that shows there is no common beta diversity patterning associated with once and twice burned groups. Even when the null deviation values of a group differ from zero, this does not mean that stochastic processes are not shaping the communities but rather there is an increasing importance of deterministic processes. To ensure that this interpretation is clear, we have added a few additional sentences that describe the results with a bit more detail. We had also neglected to note for readers that positive values of null-deviation indicate communities that are less similar than expected by chance (often inferred to point toward competition as a deterministic filter), while negative values indicate communities that are more similar than expected by chance (often inferred to point toward niche-matching or facilitation as the deterministic process). Because these interpretations reinforce the meta-community model findings we report just after this section, we’ve included a brief description of these patterns. These changes are on lines 302 – 315.

The patterns presented in figure 1 are interesting at least from the “eye ball test”. First, I think that including fire severity should be removed. It just complicates the figures. Second, I think it would be interesting to test these patterns for each fire group. I think this might provide a more interesting examination of what is happening. Take for instance rock cover for DM+LC, there certainly is a positive relationship, but I do not think there is for the other burn categories. Is there something different about DM+LC? Similarly, for No. of Logs, taxon dissimilarity seems positively correlated for CG, but not for the other sites? Is there something about CG, or is this an important factor influencing arthropods across all your sites?

We have removed the fire severity sizing and agree the figure is now easier to read. There is indeed a fire group effect on measures of taxon dissimilarity, as found in the best-fit permutational ANOVA model reported just before the figure, and the color-coding appears to reflect it. Fire group was retained in this model to deal with the structure introduced into the residuals. Fire group was not retained in the best-fit model for Shannon diversity despite the slight clumping apparent in the DM + LC group, as the remaining groups have a reasonable spread. We include these figures to help readers visualize these simple linear relationships in light of the more complicated mixed-models and permutational models that initially describe the results.

While these figures are imperfect given the more complicated models, we have used this topic you have raised as a spot to initially introduce the point about possible influences of pre-existing heterogeneity in the fire group locations—a result already reported for vegetation in these same locations (see reference 51). This occurs on lines 223 – 229. We feel that your comment greatly improved this section and added a much-needed caveat/discussion here and again in the conclusions.

Finally, the conclusion needs to be re-thought. The statement on lines 340-344 is of course possible, but if these variables are known to be key, why were they not measured?

These variables require time-intensive repeated surveys (in the case of vegetation-arthropod feedback) or expensive sensor networks (in the case of micro-climate), and these logistical challenges prevent many studies, including ours, from measuring their influence. For example, placing just one Hobo temp logger at each trapping location would cost > $6,000. They are mentioned in light of previous work that has found their influences to shape arthropod communities.

Rethink if stocasiticity is influencing arthropods in two and not all sites. Also, any statements on biological filters are just conjecture. 

We argue that stochastic influences are stronger in two of the four sites, but not absent in the remaining two. To argue that stochastic influences are not present, we would need either an environmental model that explains a very large amount of the variance in taxon patterns or a meta-community/null model combination that collectively invoke a near fixation of niche influences. We try to make this balance clear in our conclusion by highlighting the apparent influence of stochastic processes but noting that environmental factors, particularly some unmeasured values, can still be important for these communities, but we have now made it clear that these are a possible influence, not ones we measured. While we agree with your comment, for most general audiences, the deterministic view of community assembly is commonly the most popular despite longstanding evidence that stochastic assembly processes are of high importance—indeed, The Equilibrium Theory of Island Biogeography is a ‘trait-free’ probabilistic view of community assembly that greatly altered the way communities were studied. For many readers, trying to promote a more balanced view is an important step toward getting stochastic assembly processes to be considered with equal effort as deterministic.

In general, I think that the interesting aspects of this paper are that there are few predictors for arthropod community structure following disturbances. It seems that stochastic processes are probably the best model for arthropod community structure following a disturbance (although differences in arthropod community structure prior to the disturbance may also be important but unexplored in this study). Overall, your results highlight the complexities in predicting and understanding arthropod diversity across a disturbed landscape.

We agree with this central point as a take-home message and have made changes in our conclusion to better reflect this message.

Reviewer 3 Report

The study uses arthropod diversity sampling in areas burned by either one recent fire (16-20 years ago) or by two recent fires (16-20 years ago, and again 5 years ago) to assess predictions about community assembly processes following disturbance. They predicted greater influence of stochastic processes in the sites burned twice, compared to deterministic and nice-based processes in the sites burned only once, but did not find support for this prediction. Detailed analyses of the arthropod communities and site understory are provided. The manuscript is well written, with a fair discussion of the strengths and limitations of the study. I do not have direct experience with some of the analyses performed but found the methods to be described in sufficient detail. I think the work will be of interest to general ecologists due to the broad framing of the question, and to entomologists and people interested in effects of wildfires.

Minor comments:

L57-62 could provide some examples or more definition of 'deterministic processes' and 'niche-based processes' for non-ecologists who might be interested in this paper

L93-99 this is an important sentence but is written in a way that makes it unclear which communities are predicted to have the greater influence of niche-based processes. Consider rewriting and possibly breaking into multiple sentences.

L123 should be 'Table S1'? I assume this is standard practise for pitfall sampling, can you provide a general reference?

Figure 1 - Is something not quite right with the Fire Severity scale? Some of the points in the plots seem to be smaller than the legend level 0. Are there units for the Fire Severity scale? I had trouble working out what the scale meant from the methods text.

Figure 1 - There is a lot going on here, such that I have trouble extracting the key messages. One idea is to change the label color coding to make the sites that burned twice a similar colour and those that burned once a similar colour (since that is more important for the central hypothesis than the actual sites, right?... although it's nice how the colours match throughout all the figures). Was fire severity important in these models? (It's hard for me to work that out from the results text)

Table S2. State what the filled/empty boxes mean so the reader doesn't have to work it out. Some 'abundance' have a capital letter by autocorrect typo.

L350 - to me it indicates that the processes governing arthropod community assembly are highly context dependent. Or maybe many, many sites would have to be studied to see a pattern.

Author Response

L57-62 could provide some examples or more definition of 'deterministic processes' and 'niche-based processes' for non-ecologists who might be interested in this paper

We have rephrased this sentence to include more examples; the sentence now occupies lines 58 to 60.

L93-99 this is an important sentence but is written in a way that makes it unclear which communities are predicted to have the greater influence of niche-based processes. Consider rewriting and possibly breaking into multiple sentences.

We have split this long sentence into two and reworded slightly to make the prediction clearer. The sentences now fall on lines 94 to 100.

L123 should be 'Table S1'? I assume this is standard practise for pitfall sampling, can you provide a general reference?

We have corrected the table number and now included two references in the methods that describe common issues with pitfall trapping:

Querner, P., & Bruckner, A. (2010). Combining pitfall traps and soil samples to collect Collembola for site scale biodiversity assessments. Applied soil ecology, 45(3), 293-297

Woodcock, B. A. (2005). Pitfall trapping in ecological studies. Insect sampling in forest ecosystems, 37-57

Figure 1 - Is something not quite right with the Fire Severity scale? Some of the points in the plots seem to be smaller than the legend level 0. Are there units for the Fire Severity scale? I had trouble working out what the scale meant from the methods text.

Figure 1 - There is a lot going on here, such that I have trouble extracting the key messages. One idea is to change the label color coding to make the sites that burned twice a similar colour and those that burned once a similar colour (since that is more important for the central hypothesis than the actual sites, right?... although it's nice how the colours match throughout all the figures). Was fire severity important in these models? (It's hard for me to work that out from the results text)

We have taken the suggestion to remove fire severity in the marker sizes. We agree that it was complicating the figure and it is now easier to read and we think that the group color-coding is now easier to interpret as a result of removing the fire severity size variation. The sizes had an error in them, as you have noted, but with the removal of the marker size variation, this is no longer an issue.

Table S2. State what the filled/empty boxes mean so the reader doesn't have to work it out. Some 'abundance' have a capital letter by autocorrect typo.

The typos have been corrected (and Excel has been cursed-out yet again) and the shading of the boxes is now described by a column header and a short explanation in the footer.

L350 - to me it indicates that the processes governing arthropod community assembly are highly context dependent. Or maybe many, many sites would have to be studied to see a pattern.

We agree this could be the case. Changes in our discussion section now describe some of the alternative potential drivers and we conclude by encouraging more consideration of these processes.

Round 2

Reviewer 2 Report

Overall, I think you have addressed my comments. I think that we could quibble over some of the details, but the point of science is to communicate your research findings and conclusions to begin the conversation. Thank you for your efforts.